# Development of an HPLC-MS/MS Method for the Determination of Silybin in Human Plasma, Urine and Breast Tissue

**DOI:** 10.3390/molecules25122918

**Published:** 2020-06-24

**Authors:** Matteo Lazzeroni, Giovanna Petrangolini, José Antonio Legarreta Iriberri, Jaume Pascual Avellana, Digna Tost Robusté, Sara Cagnacci, Debora Macis, Valentina Aristarco, Bernardo Bonanni, Paolo Morazzoni, Harriet Johansson, Antonella Riva

**Affiliations:** 1IEO-European Institute of Oncology, Division of Cancer Prevention and Genetics, IRCCS, Via Ripamonti 435, 20141 Milan, Italy; matteo.lazzeroni@ieo.it (M.L.); sara.cagnacci@ieo.it (S.C.); debora.macis@ieo.it (D.M.); valentina.aristarco@ieo.it (V.A.); bernardo.bonanni@ieo.it (B.B.); harriet.johansson@ieo.it (H.J.); 2Research and Development Department, Indena S.p.A., Viale Ortles 12, 20139 Milan, Italy; scientificadvisor.pm@indena.com (P.M.); antonella.riva@indena.com (A.R.); 3Bioanalysis Department, Kymos Pharma Services S.L., Ronda Can Fatjó 7-B, 08290 Cerdanyola del Vallès, Spain; jlegarreta@kymos.es (J.A.L.I.); jpascual@kymos.es (J.P.A.); dtost@kymos.es (D.T.R.)

**Keywords:** silybin, silibinin, Siliphos^®^, breast cancer, HPLC-MS/MS method

## Abstract

Silybin is a flavonolignan extracted from *Silybum marianum* with chemopreventive activity against various cancers, including breast. This study was designed to develop an HPLC-MS/MS method for the determination of silybin in human plasma, urine and breast tissue in early breast cancer patients undergoing Siliphos^®^ supplementation, an oral silybin-phosphatidylcholine complex. The determination of silybin was carried out by liquid–liquid extraction with methyl-tert-butyl ether (MTBE); total silybin concentration was determined by treating the samples with β–glucuronidase, while for the determination of free silybin, the hydrolytic step was omitted. Naringenin and naproxen were selected as internal standards. The detection of the analyte was carried out by mass spectrometry and by chromatography. The HPLC-MS/MS method was evaluated in terms of selectivity, linearity, limit of quantification, precision and accuracy, and carryover. The method proved to be selective, linear, precise and accurate for the determination of silybin. To the best of our knowledge, this presents the first analytical method with the capacity to quantify the major bioactive components of milk thistle in three different biological matrices with a lower limit of quantification of 0.5 ng/mL for plasma. Silybin phosphatidylcholine, taken orally, can deliver high blood concentrations of silybin, which selectively accumulates in breast tumor tissue.

## 1. Introduction

The incidence of cancer and other degenerative disorders continues to rise in an increasingly aging population. Early diagnosis of individuals at risk of cancer has rendered chemopreventive intervention strategies highly recommended. Dietary phytochemicals are considered to be a major source of novel and safe potential cancer chemopreventive agents, mostly due to their long treatment regimen [1,2]. Silybin, also known as silibinin, is a natural polyphenolic flavonoid and the major constituent of silymarin, the standardized extract isolated from milk thistle (*Silybum marianum*), a plant traditionally used for the treatment of liver diseases [3]. 

Silymarin exerts hepatoprotective, anti-inflammatory and anti-fibrotic effects [4]. Furthermore, several pre-clinical studies have shown anti-tumoral activity of its main active component silybin in different cancer cell lines [5], as it can induce growth inhibition and apoptosis, and can enhance the therapeutic potential of doxorubicin, cisplatin, and carboplatin [6]. Most of the antitumor activity of silybin has been discovered through in vitro studies in tumor cell lines [7,8]. Fewer studies have investigated the antitumor effect exerted by in vivo supplementation with silybin on mice previously treated with carcinogens or on nude mice bearing human xenografts [9,10,11,12,13]. Some investigations have reported the effect of silymarin in the prevention and treatment of liver diseases and primary liver cancer in in vitro and in vivo models [4]. Recent data in animals reported the role of silybin in the chemoprevention of several cancer types, including breast [14,15]. 

The antitumor properties of silybin have been investigated in breast cancer mainly preclinically. Provinciali et al. studied the effect of a complex of silybin with phosphatidylcholine, Silipide^®^, which confers a higher bioavailability compared to silymarin, as previously shown in healthy human subjects [16], on the development of mammary tumors appearing spontaneously in HER-2/neu transgenic mice. The results showed marked antitumor effects, with a delay in the development of tumors and their potential metastasization. The possible mechanism of action reported was a silybin-induced down-regulation of HER-2/neu expression and a senescent-like growth arrest through a p-53 mediated pathway [15]. Despite the potential of silybin use as chemopreventive or antitumor agent to improve clinical benefits for breast cancer patients, there is still a limited knowledge of the processes regulating the absorption of silybin (and/or its metabolites) into the body, and on how it reaches its biological target. The elucidation of the mechanisms of action and the confirmation of active ingredients bioavailability are necessary to subsequently set up a chemoprevention trial in healthy subjects at high risk of cancer. 

The aim of this study was to develop an HPLC-MS/MS method for the determination of silybin in human plasma, urine and breast tissue (neoplastic or surrounding normal tissue) in early breast cancer patients undergoing Siliphos^®^ supplementation [17]. We demonstrated that Silybin phosphatidylcholine, taken orally, delivered high blood concentrations of silybin, which selectively accumulates in breast tumor tissue. The method was designed in order to determine both the free analyte (SIL) and total silybin (TOT-SIL) concentrations, taking into account the phase II metabolism of silybin (sulfate and glucuronide conjugation) [18,19,20,21,22,23].

## 2. Results and Discussion

### 2.1. Analytical Method Development

The extraction process was developed to determine both the free and total silybin concentration. Total silybin concentration (TOT-SIL, free and conjugated) was determined by treating the samples with β–glucoronidase, while for the determination of free silybin, the hydrolization phase was omitted. A liquid–liquid extraction with methyl tert-butyl ether (MTBE) was selected as a mild purification process that gave relatively clean extracts. The deconjugation process was carried out at pH 5 (also mild conditions), and the liquid–liquid extraction was not affected by the enzymatic treatment provided that only the deconjugated (and free) form of silybin (SIL) was extracted in the process, given that the conjugated forms are more hydrophilic.

Naringenin was selected as internal standard for plasma and pig muscle analysis, while naproxen was selected for urine analysis. Initially, both naringenin and apigenin were tested as possible internal standards for plasma analysis. Both of them were observed in negative ion mode, at *m/z* 270.8 and 268.7, respectively. The daughter ion peaks at *m/z* 151.0 and 117.1 were detected as the most intensive for each substance. At first apigenin was selected as internal standard, since at the chromatographic conditions established, naringenin was eluted at the same retention time of silybin. However, the experimental results demonstrated that naringenin behavior was more similar to silybin than apigenin, and the former was therefore chosen as internal standard. In human urine analysis, naproxene was used as internal standard since naringenin was present in blank urine samples at levels higher than the expected limit of quantification (LLOQ) of 1 ng/mL.

Due to the polyphenolic structure of silybin, the detection of the analyte by mass spectrometry was carried out in negative electrospray ionization using a TurboIonSpray probe. In the full scan mass spectra, the deprotonated molecular ion [M–H] of silybin was stable and exhibited higher abundance at a mass of 481.0 amu. Its ionization spectrum showed product ions at *m/z* 301 and 125.1, according to the literature [23], as the result of the break of the benzodioxin ring. The fragmentation *m/z* 481.0→301 was used for the quantification of silybin in all the samples, as it exhibited enough response for the limit of quantification established and, in addition, the resultant chromatograms had very low background with no interfering peaks. Conversely, the transition 481.0→125.1 resulted in chromatograms with many interfering peaks.

The chromatographic method was developed to determine Silybin racemic (mixture of A and B), since the reverse phase chromatographic conditions of the method were unable to establish the separation between the two diastereoisomers. Different stationary phases and mobile phases were tested considering the polyphenolic structure of the analyte in order to obtain good peak shapes and short-time runs. 

### 2.2. Method Validation

The reliability and reproducibility of the method developed were evaluated by assessing some validation parameters such as selectivity, LLOQ, calibration curve (linearity), carryover and intra-assay precision and accuracy at four concentration levels (LLOQ, low, mid and high level). The validation of the bioanalytical method consisted of one intra-day.

#### 2.2.1. Selectivity

The selectivity of plasma control, blank urine and blank pig muscle used for the preparation of calibration standards and quality control samples was assessed prior to the analysis, as silybin and internal standards can be endogenously found. This evaluation was performed after the hydrolization of the samples, since the major part of silybin is present as conjugate. Of note, while urine and pig muscle samples were analyzed only after incubation with β-glucoronidase from *H. pomatia*, blank of human plasma samples were initially processed with three different enzymes: β-glucoronidase from *H. pomatia,* β-glucoronidase from bovine liver and sulfatase from *H pomatia*. The hydrolization with β-glucoronidase from *H. pomatia* was eventually selected as the best deconjugation process.

Plasma samples of four volunteers were tested to verify the potential presence of silybin and narigenin. The plasma of one volunteer was selected as control plasma as the analyte contamination in plasma blanks was < 20% of the response of the analyte in the LLOQ, in line with FDA and EMA guidance [24,25]. Of note, FDA and EMA acceptance limits also require that the level of internal standard contamination should be < 5% of the response of the internal standard in the zero sample. In this particular case the presence of naringenin in the control plasma of the volunteer was significant, but the accuracy and precision results of the intra-assay analysis fulfilled the acceptance criteria, thus suggesting that the contribution of endogenous naringerin does not affect the correct quantification of silybin in plasma samples. 

The amount of control plasma volume was set to 200 μL, so that the sensitivity of the method reached a LLOQ of 0.5 ng/mL, and a linear range was observed between 0.5 to 500 ng/mL. Representative MRM chromatograms of blank (with no significant presence of endogenous silybin and naringenin) and plasma samples from the selected volunteer without enzyme incubation or after hydrolysis with β-glucuronidase from *Helix pomatia* are reported in Figure 1.

After the evaluation of selectivity in human plasma, the same analyses were performed on human urine from six different volunteers and on pig muscle tissue from six different animals.

For human urine analysis, a sample volume of 100 μL was selected to have a LLOQ of 1 ng/mL and a linear range between 1 to 1000 ng/mL. No significant interference peaks at the retention time of silybin nor at the retention time of naproxen were detected in the MRM chromatograms of the blank and zero sample. Appendix A shows two representative MRM chromatograms of silybin and internal standard of a human urine blank sample and a human urine zero sample. In pig muscle analysis a tissue weight of 50 mg was selected to have a LLOQ of 2 ng/g and a linear range from 2 to 500 ng/g. No significant interference peaks at the retention time of silybin nor at the retention time of naringenin were detected in the MRM chromatograms of the blank pig muscle tissues and in the zero sample. Of note, the selectivity of naringenin was verified in two different human breast tissue samples, demonstrating its absence. Two representative MRM chromatograms of silybin and internal standard of a blank sample and zero sample prepared in pig muscle tissue are presented in Appendix A.

#### 2.2.2. Calibration Curve

Calibration curves were studied at the following ranges: 0.5–500 ng/mL for human plasma, 1–1000 ng/mL for human urine, and 2–500 ng/g for pig muscle tissue. Peak area ratios of the calibration standards were fitted according to a linear model. The calibration standards were analyzed in duplicate at the beginning and at the end of the sequence. In human plasma analysis the curve plot had correlation coefficient 0.9992, slope 0.01030 and Y-intercept 0.000061. The same parameters in the human urine analysis were 0.9969, 0.00502 and −0.000909, while for pig muscle, the curve had correlation coefficient 0.9980, slope 0.00174 and Y-intercept −0.000187. The concentrations of the samples corresponding to calibration standards were back-calculated to evaluate the curve fit; the accuracies of the curve fit, expressed as mean relative error of two replicates, are reported in Table 1.

#### 2.2.3. Limit of Quantification

The LLOQ for silybin was set at 0.5 ng/mL in human plasma, 1 ng/mL in human urine and 2 ng/g in pig muscle tissue. The intra-assay accuracy (*n* = 5) at LLOQ concentration, expressed as percentage deviation in relation to the nominal concentration (relative error), was 18.40% in human plasma, 19.64% in human urine and −0.64% in pig muscle. The intra-assay precision at LLOQ, expressed as the coefficient of variation, was 13.61%, 5.14% and 9.60%, in the three analyses, respectively (Table 2). 

#### 2.2.4. Intra-Assay Precision and Accuracy

The intra-assay precision and accuracy results for the low, medium and high concentration levels (PE3–PE1) in human plasma, human urine and pig muscle are reported in Table 2. The mean intra-assay accuracies, expressed as relative error, ranged between 1.21 and 12.54% in human plasma, between −8.88 and 8.44% in human urine, and between −1.29 and 9.71% in pig muscle. The mean intra-assay precision, expressed as coefficients of variation of the calculated concentrations, were not higher than 6.38% in human plasma, 3.07% in human urine and 7.89% in pig muscle.

#### 2.2.5. Carryover

The carryover effect of the chromatographic method was performed by injecting the blank sample after the analysis of the upper limit of the calibration curve prepared in human plasma, human urine and pig muscle. The carryover effect for silybin and the internal standard was found to be not significant.

## 3. Material and Methods

### 3.1. Description of Analytical Method

The determination of silybin was carried out by liquid–liquid extraction with MTBE. Blank human plasma containing lithium heparin as anticoagulant supplied by Hospital Sant Pau, Barcelona and blank human urine were used as control samples for the preparation of the calibration curves and quality control samples for the determination of silybin. Due to the difficulty of obtaining blank breast tissue from laboratory animals, and following a review of in-house matrices available, pig muscle was chosen as a suitable surrogate matrix for the determination of silybin in human breast samples. The matrix proved to be adequate for the determination of the free and total silybin in the study samples.

#### 3.1.1. Working Solutions and Internal Standards

The test substance was silibinin (also called silybin, Sigma-Aldrich); naringenin (Sigma-Aldrich) was used as internal standard for the analyses in human plasma and breast tissue, while naproxen (Sigma-Aldrich) was used as internal standard for human urine analysis. Different working solutions of silybin and of internal standards were prepared starting from the stock solutions 1 mg/mL. Stock solutions were prepared in methanol, working solutions were prepared in methanol-water (1:1, *v*:*v*). The ranges of the working solutions were 5 ng/mL–5000 ng/mL for human plasma, 5 ng/mL–5000 ng/mL for human urine, and 5 ng/mL–1250 ng/mL for pig muscle. Working solutions were stored at −25 ± 5 °C immediately after preparation.

#### 3.1.2. Calibration Standards, Quality Control Samples, Blank and Zero Samples

Calibration standards (used for the calibration curve determination), quality control samples (used for the determination of quantitation limit, precision, and accuracy), blank (free of the analytes of interest or internal standard) and zero samples (blank samples processed with internal standard) were prepared by mixing: 20 μL of the corresponding working solutions of compound silybin, 40 μL of β-glucuronidase from *Helix pomatia* solution, 1 mL of sodium acetate 1 mol pH 5, and 200 μL of control human plasma (or 100 μL of blank human urine, or 50 mg of blank pig muscle). 

Calibration standards (PL1–PL8) were used for the calibration curve determination; the nominal concentrations of silybin in these samples in plasma, urine, and tissue are listed in Table 3. Each sample was analyzed in duplicate at the beginning and at the end of each analytical run. Quality control samples (PQ, PE3-PE1) were used for the determination of quantitation limit, precision, and accuracy. The nominal concentrations of the quality control samples are listed in Table 4. For human plasma analysis, two additional blank samples were prepared for selectivity test. One containing 20 μL of β-glucuronidase from bovine liver solution instead of the β-glucuronidase from *Helix pomatia* solution, the other containing 40 μL of sulfatase from *Helix pomatia* solution. 

#### 3.1.3. Sample Preparation

Human plasma control (0.2 mL), blank human urine (0.1 mL) and blank pig muscle (50 mg) samples were centrifuged at 3500 rpm during 10 min at 20 °C. Calibration standards, quality control samples, blank and zero samples, prepared as described above, were vortex-mixed and incubated at 37 °C for 2 h with agitation. After incubation, 20 μL of internal standard solution was added to calibration standards, quality control and zero samples, while 20 μL of methanol-water (1:1, v:v) solution was added to the blank samples to compensate the volume of internal standard solution. 

Liquid–liquid extraction was performed by adding to the samples 0.5 mL of 0.1 mol NaH_2_P0_4_ pH 2, followed by 3 mL of MTBE; the samples were then mixed for 15 minutes in a rotary shaker at room temperature. 

After liquid–liquid extraction, the samples were centrifuged for 10 min at 3750 rpm at 20 °C. The supernatant was transferred to a new tube and samples were evaporated to dryness at 45 °C by nitrogen steam; the residue was then reconstituted by adding of 0.1% acetic solution in ACN/MeOH/H20 15/35/50 (v:v:v) (1000 μL for plasma, 150 μL for urine, and 100 μL for tissue). The samples were vortex-mixed, transferred to a plastic vial, and centrifuged at 20 °C for 10 min at 3750 rpm prior to HPLC-MS/MS injection. The laboratory method for sample preparation was developed by Kymos pharma services.

#### 3.1.4. Chromatographic Method

##### HPLC Method

HPLC analyses were performed using an Agilent 1100 (plasma and tissue) or Agilent 1200 (urine) system equipped with a binary pump, a vacuum degasser, a control module system and an autosampler. To ensure the selectivity of the method for each matrix (plasma, urine or tissue), different C18 columns types and dimensions or particle size were used. The chromatographic separation was performed using a Kromasil Eternity column C_18_ (2.1 × 50 mm, 5 μm, Teknokroma) for plasma, a XBridge C_18_ column (2.1 × 100 mm, 3.5 μm, Waters) for urine and a Symmetry^®^ C_18_ column (2.1 × 50 mm, 3.5 μm, Waters) for pig muscle. Modifications to the chromatographic conditions were needed for each matrix to have a low background noise without interfering peaks at the retention time of Sibylin. The column temperature was set at room temperature. 

The solvent system consisted of a mixture of water with 0.1% acetic acid (solvent A) and acetonitrile with 0.1% acetic acid (solvent B) in different proportions according to the gradient program reported in Table 5. The flow rate was 0.5 mL/min, 0.3 mL/min and 0.4 mL/min for plasma, urine and pig muscle, respectively. The injection volume was 20 μL for plasma and pig muscle and 40 μL for urine. 

##### Mass Spectrometer

Mass spectrometry analyses were performed using an API4000 (plasma), API 3200 (urine), API3000 (pig muscle) system (MDS Sciex), equipped with a TurboIonSpray Iron Source. The system was operated in negative ionization mode with multiple reaction monitoring (MRM). The main working parameters in the analysis of human plasma were: duration time: 6 min, Gas 1 (nebulizer gas) pressure: 30 psi, Gas 2 (heater gas) pressure: 40 psi, curtain gas (CUR): 10, collision activated dissociation (CAD): 6, ion spray voltage (IS): −4200, source temperature (TEM): 550 °C. In the human urine analysis, the following working parameters were used: duration time: 9 min, Gas 1: 60 psi, Gas 2: 70 psi, CUR: 30, CAD: 5, IS: −4500 and TEM: 550 °C. In the pig muscle analysis, the working parameters were as follows: duration time: 7.5 min, Gas 1: 12, CUR: 8, CAD: 9, IS: −4000 and TEM: 500 °C. The instrument parameters for monitoring silybin and internal standards in plasma, urine and pig muscle are reported in Table 6.

### 3.2. Method Validation

The HPLC-MS/MS method was evaluated in terms of selectivity, linearity, limit of quantification, precision, accuracy, and carryover. The validation of the bioanalytical method consisted of one intra-day.

#### 3.2.1. Selectivity

The selectivity of the method was tested by comparing the chromatograms of a blank sample, a zero sample and test samples spiked with test substance at the concentration of the quantitation limit (PQ, that is QC samples prepared at the LLOQ concentration). For human plasma and human urine, a test sample spiked with the test substance at the concentration corresponding to the upper limit of the calibration curve and processed without internal standard (PL1n) was also tested. The chromatograms of the samples were compared to evaluate the potential interferences with the peaks of silybin and the internal standard. 

The following acceptance criteria were applied:Area response of the potential peak eluting with the same retention time, molecular weight and fragmentation transition as silybin <20% of the corresponding response of this compound in the lower limit of quantification (LLOQ).Area response of the potential peaks eluting with the same retention time, molecular weight and fragmentation transition as internal standard <5% of the corresponding response of this compound at the concentration used in the study.

#### 3.2.2. Calibration Curve

Calibration curves were evaluated over the concentration range for silybin using the calibration standards described in Section 3.1.1. At least two blank and zero samples were prepared and analyzed together with the calibration curve and each analysis was performed in duplicate. The area response ratio (silybin vs. internal standard) was fitted to the nominal concentration using the simplest model through Analyst software (version 1.4.1 for plasma, 1.4.2 for urine and 1.4.2 for tissue).

The following acceptance criteria were applied:Correlation coefficient ≥ 0.99.Accuracy of the back-calculated concentrations ≤ 15% for all concentration levelsAccuracy of the LLOQ ≤ 20%.

At least 75% standards with a minimum of six calibration standard levels should meet the above criteria, including the LLOQ and the highest concentration standard. Calibration standards that did not achieve this criterion were dropped.

#### 3.2.3. Limit of Quantification

The quantitation limit was evaluated using the quality control sample PQ. Five replicates were prepared and analyzed in one batch. The concentrations of silybin at LLOQ were determined by interpolation from the calibration curve using Analyst software (version 1.4.1 for plasma, 1.4.2 for urine and 1.4.2 for tissue).

The following acceptance criteria were applied:Intra-assay precision: coefficient of variation of the LLOQ concentrations in the five replicates ≤ 20%Intra-assay accuracy: mean percentage deviation of the concentrations of the five replicates within ± 20% of the nominal value.

#### 3.2.4. Determination of Precision and Accuracy

The intra-assay precision and accuracy were determined at low, medium and high concentration levels. We used the quality control samples (PE1, PE2, PE3), analyzing five replicates for each concentration level. The concentration of silybin in samples was determined by interpolation from the calibration curve using Analyst software (version 1.4.1 for plasma, 1.4.2 for urine and 1.4.2 for tissue). Accuracy was expressed as a percentage deviation of the calculated concentration from the nominal concentration. Precision was determined from the calculated concentrations of the replicates of each concentration level by using the coefficient of variation.

The following acceptance criteria were applied:Intra-assay precision: coefficient of variation of the analyte concentrations in the five replicates ≤ 15%Intra-assay accuracy: mean percentage deviation of the concentrations of the five replicates within ± 15% of the nominal value.

#### 3.2.5. Carryover

Carryover effects were evaluated by injecting a blank after one sample corresponding to the upper limit of the calibration curve in the intra-assay validation batch. Acceptance criteria were the same described in Section 3.2.1 (Selectivity).

## 4. Conclusions

An HPLC-MS/MS method was developed for the determination of silybin in human plasma, urine and breast tissue. The method consisted of an enzymatic hydrolysis of the samples with β-glucoronidase of *Helix pomatia* followed by a liquid–liquid extraction with MTBE using naringenin (plasma and pig tissue) or naproxen (urine) as internal standards. The LLOQ was set at 0.5 ng/mL for human plasma, 1 ng/mL for human urine and 2 ng/g for pig muscle tissue. The concentration ranges of the method, in which a linear fitting model was applied, was set from the LLOQ to the highest concentration level (500 ng/mL in human plasma, 1000 ng/mL in human urine and 500 ng/g in pig muscle tissue). The method proved to be selective, linear, precise and accurate for the determination of silybin. To the best of our knowledge, this presents the first analytical method with the capacity to quantify the major bioactive components of milk thistle in three different biological matrices. Furthermore, with regard to plasma quantification of silybin, our method provided a much lower limit of quantification than that reported in the literature by Wen Z et al. (0.5 ng/mL vs. 2 ng/mL) [23]. This method was found to be readily applicable to the clinical study (*Code n. R621-IEO661/511*), where for the first time in early breast cancer patients, the breast tissue distribution of silybin and its effect on cell proliferation and other biomarkers were determined [17]. The authors showed for the first time that oral silybin-phosphatidylcholine reaches biologically relevant breast tumor tissue concentrations (TOT-SIL up to 1375 ng/g, SIL up to 177 ng/g), in correlation with its high blood concentrations (TOT-SIL ranged from 31,121 to 7654 ng/mL and SIL ranged from 10,861 to 1818 ng/mL) of silybin (Appendix A). These results provide the basis for future clinical studies of Siliphos^®^ in breast cancer prevention.

## Figures and Tables

**Figure 1 molecules-25-02918-f001:**
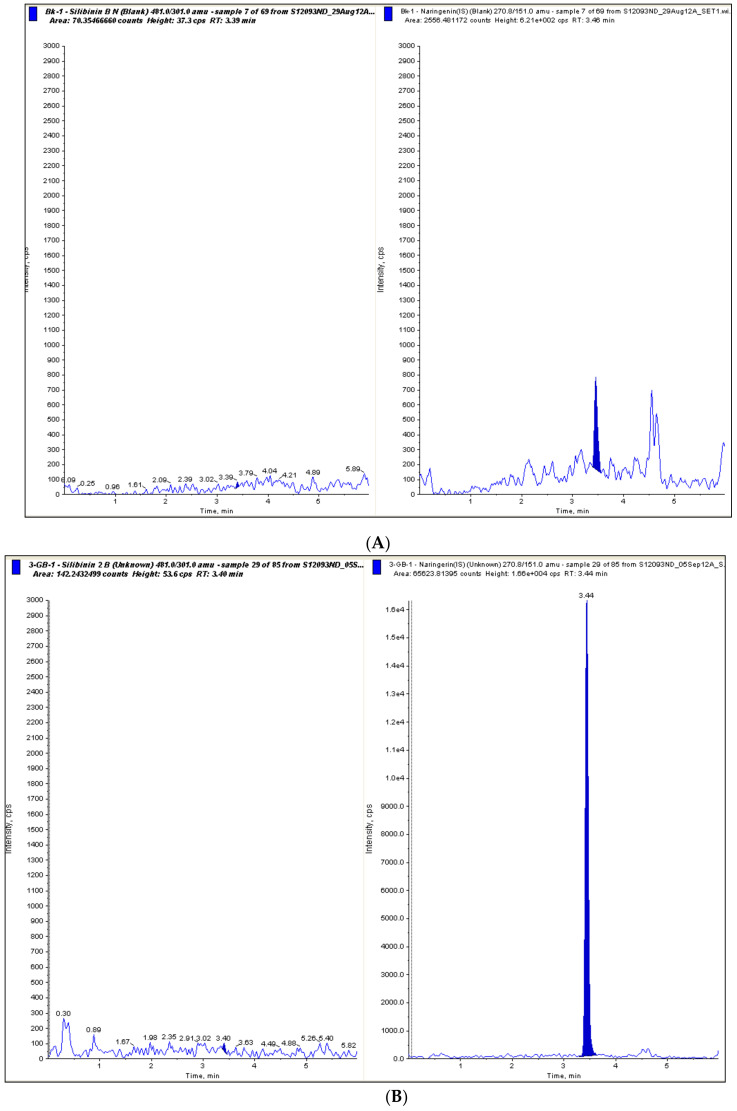
MRM chromatogram corresponding to a plasma blank sample without enzyme incubation (**A**) and a plasma blank sample incubated with β-glucuronidase from *H. pomatia* (**B**). In each panel: on the left, MRM chromatogram corresponding to silibinin; on the right, MRM chromatogram corresponding to the internal standard.

**Table 1 molecules-25-02918-t001:** Accuracy of the back-calculation concentrations on two samples in human plasma, human urine, and pig muscle analyses.

Calibration Sample	Accuracy (RE, %)		
	Human Plasma	Human Urine	Pig Muscle
PL1	−1.87	−11.54	−1.18
PL2	−0.83	−5.25	−2.12
PL3	1.30	−0.18	3.08
PL4	−0.18	1.84	1.80
PL5	1.03	4.57	6.26
PL6	2.29	3.32	−3.42
PL7	−1.51	4.74	−2.69
PL8	0.27	−6.53	1.40

RE: Relative error.

**Table 2 molecules-25-02918-t002:** Accuracy and precision for calibration standards in human plasma, human urine, and pig muscle analyses.

Sample	Mean	RE (%)	C.V. (%)
**Human Plasma**			
PQ	0.59	18.40	13.61
PE3	1.69	12.54	6.38
PE2	16.80	12.03	1.66
PE1	404.85	1.21	3.74
**Human urine**			
PQ	1.196	19.64	5.14
PE3	3.015	0.51	3.07
PE2	32.533	8.44	1.29
PE1	728.992	−8.88	2.95
**Pig Muscle**			
PQ	1.987	−0.64	9.60
PE3	6.583	9.71	7.89
PE2	26.007	8.36	3.20
PE1	394.834	−1.29	3.76

C.V.: Coefficient of variation; RE: Relative error.

**Table 3 molecules-25-02918-t003:** Calibration standards and corresponding concentrations of silybin used in human plasma, human urine and pig muscle analyses.

Calibration Standard	Human Plasma	Human Urine	Pig Muscle
	Silybin Concentration (ng/mL)	Silybin Concentration (ng/mL)	Silybin Concentration (ng/g)
PL1	500	1000	500
PL2	250	500	250
PL3	100	200	100
PL4	50	100	50
PL5	10	20	25
PL6	2.5	5	10
PL7	1	2	5
PL8	0.5	1	2

**Table 4 molecules-25-02918-t004:** Quality control samples and silybin concentrations in human plasma, human urine, and pig muscle analyses.

Quality Control Sample	Human Plasma	Human Urine	Pig Muscle
	Silybin Concentration (ng/mL)	Silybin Concentration (ng/mL)	Silybin Concentration (ng/g)
PQ	0.5	1	2
PE3	1.5	3	6
PE2	15	30	24
PE1	400	800	400

**Table 5 molecules-25-02918-t005:** Gradient program for HPLC method for human plasma, human urine and pig muscle analyses.

Human Plasma		Human Urine		Pig Muscle	
Time (Minutes)	Mobile Phase B (%)	Time (Minutes)	Mobile Phase B (%)	Time (Minutes)	Mobile Phase B (%)
0.00	15	0.00	25	0.00	25
3.00	80	3	75	2.50	90
4.50	80	6	75	3.50	90
4.60	15	6.6	25	3.60	25
6.00	15	9	25	7.50	25

A: 0.1% acetic acid in water; B: 0.1% acetic acid in acetonitrile.

**Table 6 molecules-25-02918-t006:** Mass spectrometer parameters for monitoring silybin and internal standards in human plasma, human urine, and pig muscle analyses.

Compound	Q1 Mass	Q3 Mass	Time (msec)	DP(v)	FP (v)	EP (v)	CE (eV)	CXP(v)
Human Plasma								
Silybin	481.0	301.0	200	−120	-	−11	−28	−7
Naringenin	270.8	151.0	200	−100	-	−11	−26	−9
Human Urine								
Silybin	481.2	301.0	200	−75	-	−10	−28	−10
Naproxen	229.0	184.9	200	−23	-	−2	−11	−4
Pig Muscle								
Silybin	481.2	301.0	200	−80	−210	−10	−29	−18
Narigenin	271.0	150.9	200	−70	−150	−8	−24	−11

CE = collision energy; CXP = collision cell exit potential; DP = declustering potential; EP = entrance potential; FP = focusing potential.

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
