# Peer review of "Development of an HPLC-MS/MS Method for the Determination of Silybin in Human Plasma, Urine and Breast Tissue"

_molecules, 2020, doi:10.3390/molecules25122918_

Round 1

Reviewer 1 Report

Manuscript "Development of an HPLC-MS/MS method for the determination of siline in human plasma, urine and breast tissue" addresses the development of an analytical methodology to quantify silybin in three different matrices, plasma, urine and pig muscle, besides its validation. The work brings relevance to provide an alternative to validated silybin quantification methods, as it is more sensitive. In addition to enabling clinical studies with milk thistle.

Before accepting the manuscript, it is necessary to pay attention to the corrections suggested below:

line 18 - Silybum marianum must be in italic

throughout the manuscript, the m/z must be in italic

line 98 - it's not transition, it's fragmentation ... m/z 481 -> m/z 125

line 103 - How have you determined the mixture is racemic?

Author Response

  • line 18 - Silybum marianum must be in italic: Reply: DONE
  • throughout the manuscript, the m/z must be in italic: Reply: DONE
  • line 98 - it's not transition, it's fragmentation ... m/z 481 -> m/z 125: Reply: the sentence has been corrected.
  • line 103 - How have you determined the mixture is racemic? Reply: The reverse phase chromatographic conditions of the method are unable to resolve Silybin A and Silybin B, therefore, Silybin racemic is determined. The explanation has been written in the manuscript.

Reviewer 2 Report

Following minor revision should be done:

I think that one chromatogram is enough for exemplification and the others could be give as supplementary material.

As long as the characteristics of the calibration curves are given, it is no longer necessary to present them as figures.

What is the solvent used for preparation of working and standards solutions?

The method used for sample preparation is yours or is taken from literature? Please specify this.

Why didn't you use the same HPLC device for all chromatographic analysis?

Author Response

  • I think that one chromatogram is enough for exemplification and the others could be give as supplementary material. Reply: Ok. We have accepted the reviewer's advice.
  • As long as the characteristics of the calibration curves are given, it is no longer necessary to present them as figures. Reply: Ok. We have accepted the reviewer's advice. We left only figure 1, the others became supplementary and figure 4 has been deleted as suggested by reviewer #3.
  • What is the solvent used for preparation of working and standards solutions? Reply: Stock solutions were prepared in methanol, working solutions are prepared in methanol:water 1:1 v/v. The information has been specified into the text.
  • The method used for sample preparation is yours or is taken from literature? Please specify this. Reply: The methods for the three matrices were developed in Kymos. The information has been specified into the text.
  • Why didn't you use the same HPLC device for all chromatographic analysis? Reply: The MS equipment is chosen based in the sensitivity required for each matrix (API 4000 for plasma, API3200 for urine or API3000 for tissue). Each MS has a different HPLC attached. The reasons has been specified into the text.

Reviewer 3 Report

In this paper the LC-MS/MS method for the determination of silybin in three different biological matrices (pig muscle tissue, huma plasma and urine) was developed for the firs time. The advantage of the proposed method is much lower limit of quantification comparing to the previous studies. However to consider its publication, the minor revision should be performed. My suggestions are:

  • General remark about the manuscript is that it should be written in a more clearly way. It is difficult to understand some parts of the text.
  • ‘Natural products/Edible plants/Dietary phytochemicals (…)’, in my opinion it is enough to use dietary phytochemicals.
  • Line 107 and 301. Should be ‘Method validation’ or ‘Analytical performance characteristic of the method’.
  • Apart of LOQ values please provide also LOD values for all investigated types of samples.
  • Please calculate inter-assay precision of the method.
  • Please enlarge the font size of all figures in the manuscript.
  • Caption of Figure 4 appears 3 times in the manuscript (lines: 180, 182 and 184).
  • In my opinion it is not necessary to show Fig. 4.
  • Line 184. Please move “2.2.3. Limit of quantification” to the next line.
  • Tables 2 and 6. Please remove letters a, b and c from the tables and the titles of tables. Combine the tables marked with the letters a, b, c into one.
  • Line 214. The abbreviation of methyl tert-butyl ether (MTBE) should be introduced earlier after the first mention in the text (line 79).
  • Line 212. Section ‘3.1. Description of analytical method’. Analytical method is a specific method of determining a compound (or compounds) in a specific sample, by means of a specific technique, which is the basis of the analytical process. Therefore, sections 3.2 should be rather subsection of section 3.1.
  • Lines 222-224. Silibinin, naringenin and naproxen should be written with a lowercase letter.
  • Line 224. Should be “(…) was used as internal standard for human urine analysis.”
  • The manuscript lacks information on what solutions (water, methanol) were used to prepare stock solutions and working solutions of silybin and internal standards?
  • Lines 226-227. It should be “(…) were 5 ng/mL - 5000 ng/mL for human plasma, 5 ng/mL -5000 ng/mL for human urine and 5 ng/mL - 1250 ng/mL (…)”.
  • English should be improved, e.g. line 225 “the stock solutions”, line 226 ”the working solutions”, line 220 “in the studied samples”, other lines where corrections are necessary: 176, 239, 247, 260-261, 303, 319-320, 343-344, 357-358.
  • Lines 230-235. The solutions added should be given after the decimal points, without dashes, according to the scheme: “Calibration standards, quality control samples, blank and zero samples were prepared by mixing: 20 uL of the corresponding working solutions of silybin, 40 μL of β-glucuronidase from Helix pomatia solution, (…)”.
  • Quality control samples, blank, zero, test samples – this terms should be clearly defined in the manuscript.
  • Please use mol/L instead of M and ‘mL’ instead of ‘ml’ throughout the manuscript.
  • Line 250. What was the volume/mass of the samples?
  • Line 254. It is not clear why methanol:water (1:1) solution was added to the blank samples. Moreover, it should be written (1:1, v:v).
  • Did the Authors optimize the type and volume of the extractant and LC-MS/MS method parameters? If so, it should be described in detail.
  • Line 259. ‘(…) samples were evaporated to dryness at 45°C’. Was it evaporated by nitrogen stream? This information should be included in the manuscript.
  • Line 261. ‘ACN/MeOH/H20 15/35/50 0,1% acetic acid’ this expression is not clear. Moreover, it should be written ACN/MeOH/H20 15/35/50 (v:v:v).
  • Line 293. It is more clear when it is written: A: 0.1% acetic acid in water; B: 0.1% acetic acid in acetonitrile.
  • Lines 280-281. Use commas: ‘(…) an API4000 (plasma), API 3200 (urine), API3000 (pig muscle) systems’.
  • Lines 283-288. Please use colons, e.g. duration time: 6 min, Gas 1 (nebulizer gas) pressure: 30 psi, Gas 2 (heater gas) pressure: 40 psi, Gas 1: 60 psi, Gas 2: 70 psi, CUR: 30, CAD: 5, IS: -4500 etc.
  • Line 288. What was the Gas 2 pressure?
  • What is the difference between the quantitation limit (PQ) (line 307) and limit of quantification (LLOQ)?
  • The manuscript is missing a table that contains information aboutthe results of the determination of silybin in investigated biological samples (tissue, plasma and urine), with the determined concentration ranges, mean values, standard deviations and the number of analyzed samples.
  • Line 427. Should be ‘Application of Liquid Chromatography-electrospray (…)’.

Author Response

  • Natural products/Edible plants/Dietary phytochemicals (…)’, in my opinion it is enough to use dietary phytochemicals. Reply: we left dietary phytochemicals
  • Line 107 and 301. Should be ‘Method validation’ or ‘Analytical performance characteristic of the method’. Reply: we wrote method validation in both the lines.
  • Apart of LOQ values please provide also LOD values for all investigated types of samples. Reply: The LOD is not determined as part as the bioanalytical method. The concentration values below the LLOQ of the method are reported as BLQ (below the lower limit of quantitation).
  • Please calculate inter-assay precision of the method. Reply: The validation of the bioanalytical method consisted of one intra-day.
  • Please enlarge the font size of all figures in the manuscript. Reply: Ok, original plasma figures attached.
  • Caption of Figure 4 appears 3 times in the manuscript (lines: 180, 182 and 184). Reply: figure 4 has been deleted.
  • In my opinion it is not necessary to show Fig. 4. Reply: figure 4 has been deleted.
  • Line 184. Please move “2.2.3. Limit of quantification” to the next line. Reply: done.
  • Tables 2 and 6. Please remove letters a, b and c from the tables and the titles of tables. Combine the tables marked with the letters a, b, c into one. Reply: tables have been re-formatted.
  • Line 214. The abbreviation of methyl tert-butyl ether (MTBE) should be introduced earlier after the first mention in the text (line 79). Reply: done.
  • Line 212. Section ‘3.1. Description of analytical method’. Analytical method is a specific method of determining a compound (or compounds) in a specific sample, by means of a specific technique, which is the basis of the analytical process. Therefore, sections 3.2 should be rather subsection of section 3.1. Reply: sections have been modified according to reviewer's advices.
  • Lines 222-224. Silibinin, naringenin and naproxen should be written with a lowercase letter. Reply: done.
  • Line 224. Should be “(…) was used as internal standard for human urine analysis.” Reply: the sentence has been corrected.
  • The manuscript lacks information on what solutions (water, methanol) were used to prepare stock solutions and working solutions of silybin and internal standards? Reply: Stock solutions are prepared in methanol, working solutions are prepared in methanol:water 1:1 v/v. The information has been integrated into the text.
  • Lines 226-227. It should be “(…) were 5 ng/mL - 5000 ng/mL for human plasma, 5 ng/mL -5000 ng/mL for human urine and 5 ng/mL - 1250 ng/mL (…)”. Reply: the text has been modified.
  • English should be improved, e.g. line 225 “the stock solutions”, line 226 ”the working solutions”, line 220 “in the studied samples”, other lines where corrections are necessary: 176, 239, 247, 260-261, 303, 319-320, 343-344, 357-358. Reply: all the sentences have been checked and the english improved.
  • Lines 230-235. The solutions added should be given after the decimal points, without dashes, according to the scheme: “Calibration standards, quality control samples, blank and zero samples were prepared by mixing: 20 uL of the corresponding working solutions of silybin, 40 μL of β-glucuronidase from Helix pomatia solution, (…)”. Reply: solutions have been described using the scheme suggested by the reviewer.
  • Quality control samples, blank, zero, test samples – this terms should be clearly defined in the manuscript. Reply: The terms have been defined in the manusctipt.
  • Please use mol/L instead of M and ‘mL’ instead of ‘ml’ throughout the manuscript. Reply: ok.
  • Line 250. What was the volume/mass of the samples? Reply: Human plasma control (0.2 mL), blank human urine (0.1 mL) and blank pig muscle (50 mg). The information has been integrated into the text.
  • Line 254. It is not clear why methanol:water (1:1) solution was added to the blank samples. Moreover, it should be written (1:1, v:v). Reply: MeOH:water is added to blank plasma samples to compensate the volume of internal standard solution (prepared in MeOH:water 1:1, v:v) added to calibration standards, quality control and zero samples. The information has been integrated into the text.
  • Did the Authors optimize the type and volume of the extractant and LC-MS/MS method parameters? If so, it should be described in detail. Reply: The optimum LC-MS condition are now clearly described in section 3.1.4.
  • Line 259. ‘(…) samples were evaporated to dryness at 45°C’. Was it evaporated by nitrogen stream? This information should be included in the manuscript. The evaporation was by nitrogen stream. The information has been integrated into the text.
  • Line 261. ‘ACN/MeOH/H20 15/35/50 0,1% acetic acid’ this expression is not clear. Moreover, it should be written ACN/MeOH/H20 15/35/50 (v:v:v). Reply: The solution used is 0.1% acetic solution in ACN/MeOH/H20 15/35/50 (v:v:v). The sentence has been corrected.
  • Line 293. It is more clear when it is written: A: 0.1% acetic acid in water; B: 0.1% acetic acid in acetonitrile. Reply: ok.
  • Lines 280-281. Use commas: ‘(…) an API4000 (plasma), API 3200 (urine), API3000 (pig muscle) systems’. Reply: ok.
  • Lines 283-288. Please use colons, e.g. duration time: 6 min, Gas 1 (nebulizer gas) pressure: 30 psi, Gas 2 (heater gas) pressure: 40 psi, Gas 1: 60 psi, Gas 2: 70 psi, CUR: 30, CAD: 5, IS: -4500 etc. Reply: ok.
  • Line 288. What was the Gas 2 pressure? Reply: The tissue was analysed on an API3000 instrument that uses a different ionization source, only Gas 1 (NEB Gas) applies.
  • What is the difference between the quantitation limit (PQ) (line 307) and limit of quantification (LLOQ)? Reply: PQ is the term used for QC samples prepared at the LLOQ concentration.
  • The manuscript is missing a table that contains information about the results of the determination of silybin in investigated biological samples (tissue, plasma and urine), with the determined concentration ranges, mean values, standard deviations and the number of analyzed samples . Reply: We added a table of the results of the mail trial.Line 427. Should be ‘Application of Liquid Chromatography-electrospray (…)’. Reply: ok